# Incidence of Carpal Tunnel Syndrome and Other Coexisting Brachial Plexus Neuropathies in Bullseye Shooters—A Pilot Retrospective Clinical and Neurophysiological Assessment

Aleksander Rajczewski [1,†], Przemysław Daroszewski [2,†], Artur Fabijański [1], Ksawery Bogusławski [1], Michał Kaźmierczak [1] and Juliusz Huber [1,*]

1   Department of Pathophysiology of Locomotors Organs, University of Medical Sciences, 61-701 Poznań, Poland; aleksander.rajczewski@gmail.com (A.R.); artur.fab@gmail.com (A.F.); xboguslawski@gmail.com (K.B.); michal.kazmierczak1996@gmail.com (M.K.)
2   Department of Organization and Management in Health Care, University of Medical Sciences, 61-701 Poznań, Poland; dyrektor@orsk.ump.edu.pl
*   Correspondence: juliusz.huber@ump.edu.pl; Tel.: +48-12-504-041-843
†   These authors contributed equally to this work.

**Abstract:** Shooting may impact the frequency of neuropathies in the upper extremity nerves or of cervical disc–root conflicts. This study was undertaken to assess whether shooting sports trained with a handgun by civilians may influence the risk factor for carpal tunnel syndrome (CTS) and other neuropathies of the brachial plexus nerve fibers. Neurophysiological studies using surface electromyography (rEMG at rest and mcEMG during maximal contraction), electroneurography (ENG), and motor-evoked potential recordings (MEPs) were performed in a select population of nine shooters, which were rigorously screened as positive through a clinical examination for carpal tunnel syndrome and other brachial plexus neuropathies among a population of forty-two subjects, to confirm the existence of pathologies in the upper extremities. Increased muscle tension in rEMG and a simultaneous decrease in motor unit activity in mcEMG were recorded both in the proximal and distal muscles of the upper extremities more frequently in the shooters than in the healthy controls—volunteers. An ENG examination confirmed CTS in the shooting hand of four subjects (4/42; 9.5%), additionally revealing a significantly decreased F-wave at the C6–C7 levels in the dominant extremities of the shooting group in comparison to the control population ($p = 0.05$). All the examined subjects had revealed brachial plexus pathologies on both sides according to the results of the MEP recordings upon stimulation at the C4–C8 levels (various significant differences between the shooters and control group were found), and two had ulnar neuropathy in the wrist on the shooting side. It was concluded that shooting sports are a moderate risk factor for carpal tunnel syndrome and that they significantly influence the development of other brachial plexus neuropathies.

**Keywords:** shooting sports; brachial plexus neuropathies; carpal tunnel syndrome; clinical studies; neurophysiological assessment; surface electromyography; motor-evoked potentials

## 1. Introduction

The shooting position forces the wrist, elbow, and shoulder to be overloaded, which may cause the dysfunction of neural impulse transmission in these three areas within the anatomical passage of the nerve branches of the upper extremity [1,2]. Earlier studies on the aforementioned dysfunctions in sports disciplines, such as golf [3] and archery [4], showed abnormalities in neurophysiological studies on the functioning of the motor units of the muscles responsible for gripping and stretching. An analysis of the available literature does not provide neurophysiological evidence of similar pathologies apart from the diagnosis of radial nerve palsy caused by military shooting training [5]. It should be expected that the dysfunction of the musculoskeletal system in shooters, caused by both holding a

handgun and a recoil, may contribute to the development of peripheral neuropathies or cause damage to the nerve fibers at the level of the arm or spine as presented by Zeman and Pitr [6]. However, they did not use a precise instrumental methodology to identify such dysfunctions and focused mainly on spinal pathologies assessed through a clinical examination. It can also be assumed that a regular history of shooting affects the severity of neuropathies, such as carpal tunnel syndrome (CTS), ulnar neuropathies of the wrist or elbow, or abnormalities in brachial plexus fiber neural transmission. The knowledge gap may lie either in the low incidence of complications or in the use of low-sensitivity clinical tools to assess pathologies. Modern clinical neurophysiological tests, such as electromyography (EMG) [7,8], electroneurography (ENG) [9], and recently motor-evoked potentials (MEPs) [10], allow for a precise assessment of sensory and motor deficits; in our study, we used all of the techniques mentioned as a novum in the research. When reviewing the latest literature on the discussed problem, we did not come across publications in which the function of the sensory and motor fibers of the brachial plexus nerves in civilian and military shooters was assessed using the diagnostic tests of clinical neurophysiology used in this study. Severe brachial plexus injuries are rarely mentioned in sports medicine, the topic of the diagnosis is described differently, and their treatment requires a multidisciplinary team. The clinical examination is the milestone that defines and locates an injury. Electrodiagnostic methods and imaging techniques provide valuable morphological and functional information that can guide treatment and determine a prognosis [2]. The lack of a description in the literature of the handgun recoil consequences on the function of the muscles and nerves in the upper extremities was the main reason to undertake the problem in the presented study. Most of the clinical and neurophysiological reports were dedicated more to gunshot wounds [11].

This study was undertaken to assess with the neurophysiological methods whether shooting sports trained with a handgun by civilians are a risk factor for the development of carpal tunnel syndrome (CTS) and other neuropathies in the brachial plexus nerve fibers. The null hypothesis is that shooting does not cause peripheral mononeuropathy or predispose shooters to disc–root conflicts.

## 2. Materials and Methods

### 2.1. Participants and Study Design

Figure 1 provides a visual representation of the overall design and course of this study. We recruited shooters who met the inclusion criteria and were then screened for CTS. Those who tested positive in clinical evaluation underwent a neurophysiological examination.

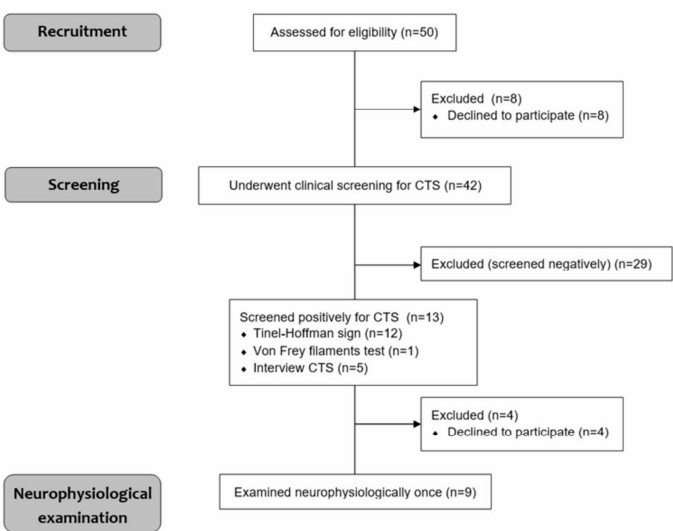

**Figure 1.** Study design.

The initial trial included 50 members of a local shooting club of both sexes. Before the study, the subjects were asked to confirm their history of systematic training with handguns (minimum of 6 months), and the lack of subjective symptoms ("no subjective symptoms") confirmed the development of pathologies in the upper extremities, such as motor and sensory impairments and cervical pain. Subjects gave written informed consent to participate in the study. Exclusion criteria included severe spinal or head trauma, history of stroke and other neurological or autoimmunological diseases, recent pregnancy, and presence of a pacemaker or cochlear implant. Eight people declined to participate. In total, 42 patients underwent a clinical evaluation for CTS, which was negative in 29 shooters and positive in 13 shooters. We used our previous experience in neurologic examination to detect brachial plexus or thoracic outlet syndrome pathologies [10]. At this stage of recruitment, four other people refused to participate in neurophysiological tests, which were aimed at precise confirmation of impairments within the peripheral nervous system. Ethical considerations were in agreement with the Declaration of Helsinki. The approval of the Bioethics Committee of the Poznań University of Medical Sciences was also obtained (including studies on healthy people, decision No. 554/17). Data on the anthropometric characteristics and shooting practice details of nine neurophysiologically tested shooters were collected (Table 1).

**Table 1.** Demographic and anthropometric properties and results of clinical studies performed on healthy subjects (N = 9) and shooters (N = 9). Median or mean values with standard deviations are presented.

| Variable | Control (N = 9) | Shooters (N = 9) | $p$ |
|---|---|---|---|
| Age (years) | $41.6 \pm 13.0$ | $43.4 \pm 14.1$ | 0.08 |
| Weight (kg) | $76.2 \pm 11.8$ | $82.4 \pm 16.1$ | 0.06 |
| Height (cm) | $172.8 \pm 6.5$ | $174.7 \pm 10.1$ | 0.08 |
| BMI | $25.4 \pm 2.9$ | $27.0 \pm 4.8$ | 0.08 |
| Shooting hand/dominant hand | Right = 9 | Right = 8 Left = 1 | 0.05 |
| Practice (years) | NA | $7.1 \pm 36.0$ | NA |
| Frequency of training (x/month) | NA | $4.8 \pm 3.4$ | NA |
| Pain VAS (0–10) | 0 | 1 | 0.06 |
| vfF filaments, 2nd fingertip (C6–C7) | 1 | 2 | 0.02 |
| vfF filaments, 5th fingertip (C8–T1) | 1 | 2 | 0.02 |
| vfF filaments, area over biceps (C5) | 1 | 2 | 0.05 |

Abbreviations: vfF filaments—von Frey filaments of superficial sensory perception studies (0—analgesia, 1—normal, 2—hyperesthesia)—and NA—nonapplicable, $p \leq 0.05$.

As a control population of subjects, we recruited a group of healthy volunteers (N = 9) that was equal to the number of shooters examined neurophysiologically. Shooters and healthy volunteers did not differ significantly in gender, age, height, weight, BMI, or dominant hand (right). They were tested with the same neurophysiological methods used on the shooters to obtain reference values of recorded parameters for comparison.

Among the clinical assessment methods, we used three tests: (1) von Frey filament tactile method (vFf) (for the detection of impairment in sensory perception [12–14]), (2) Tinel–Hoffman sign (for the detection of pathology symptoms in the median nerve [15]), (3) Froment's sign (for the detection of pathology symptoms in the ulnar nerve [1]), and (4) history of CTS-specific symptoms. After obtaining at least one positive result, the shooter was referred for neuropsychological tests.

The tactile method of von Frey filaments to assess sensory perception was used in three areas of the upper extremity covering different dermatomes (2nd finger—C6–C7, 5th finger—C8–T1, and the area above the biceps—C5). It can diagnose analgesia (score of 0), normal perception (score of 1), or hyperalgesia (score of 2).

The final trial included two women and seven men, whose anthropometric characteristics are described in Table 1. Overall, they practiced for seven years on average but no less than six months, with almost five shooting sessions per month.

*2.2. Neurophysiological Testing*

All tests were performed once with four-channel Keypoint System (Medtronic A/S, Skovlunde, Denmark) in the prone position (Figure 2A–C) according to the guidelines of European Chapter of The International Federation of Clinical Neurophysiology.

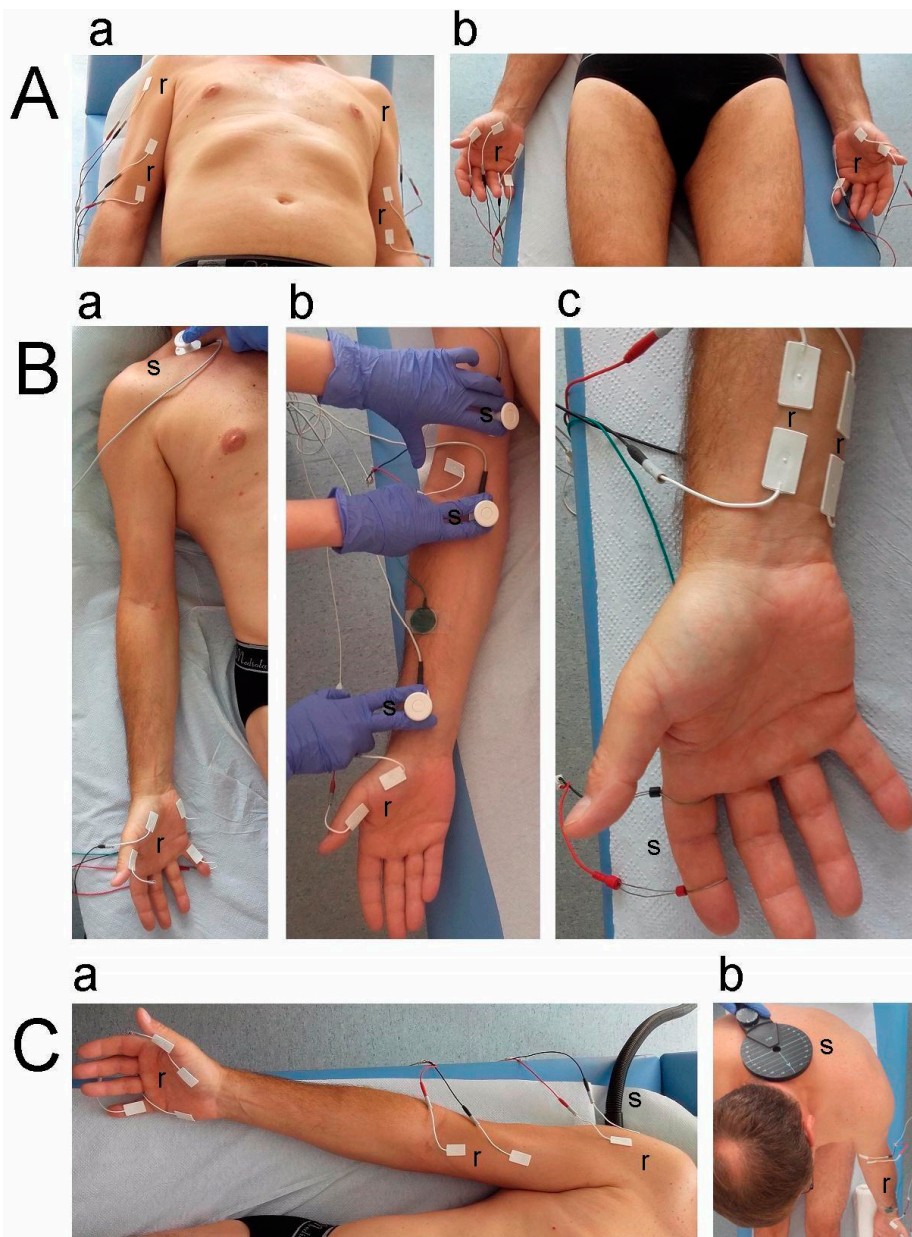

**Figure 2.** Photographs of experimental setup applied during neurophysiological recordings. (**A**) Sites of bilateral placement of bipolar surface electrodes over muscles in proximal (a) and distal (b) parts of upper extremities when sEMG recordings at rest and during maximal contraction were performed. (**B**) Sites of bilateral electrical stimulation of motor nerve fibers in the median and ulnar nerves at Erb's point (a), arm, cubital fossa or ulnar sulcus (not shown), and wrist (b) during ENG studies of M-wave recordings of hand muscles. The methodology of sensory conduction studies (SCV) (c) on the median and ulnar nerves (not shown) following stimulation of fingertips with recordings of prewrist areas. (**C**) Principles of multiple motor-evoked potentials (MEP) recordings from upper extremity muscles (a) following transvertebral magnetic stimulation of motor centers at C5–C8 spinal neuromeres (b). Abbreviations: r—recording—and s—stimulation.

Surface electromyography (sEMG, Figure 2A) was performed bilaterally at rest (rEMG) and during maximal muscle contraction (mcEMG) on the deltoideus (DEL), biceps brachii (BB), abductor digiti minimi (ADM), and abductor pollicis brevis (APB) for 5 s. Parameters of amplitudes (in µV) in rEMG and mcEMG recordings and, additionally, frequencies (in Hz) of recruiting action potentials of motor units were analyzed and ranked from normal (3) to severe pathology (1). Standard disposable Ag/AgCl surface electrodes with an active surface of 5 mm$^2$ were used. The active electrodes were placed over the belly of the examined muscle, while the reference electrodes were placed on the distal part of the tendon. The ground electrode was placed close to the recording pair. Upper 10 kHz and lower 20 Hz filters of the recorder were used. In the first stage of the study, the patient was asked to completely relax the examined muscles and then to perform a maximum contraction for 5 s during which simultaneous recording took place. Subjects were instructed to contract the tested muscle as hard and fast as possible until the neurophysiologist asked to end the test. The test was performed three times with a one-minute rest period between each muscle contraction. The recording with the highest amplitude (in µV) and frequency (in Hz) parameters was selected for the final analysis. sEMG recordings were made with a time base of 80 ms/D and an amplification of 20–1000 µV. The methodical and analytical principles of sEMG have been described elsewhere [7,8,10,16].

Electroneurography (ENG, Figure 2B) included the assessment of peripheral neural transmission in the motor and sensory fibers of the median and ulnar nerves bilaterally after stimulation with electrical impulses over the anatomical passage in the prewrist, cubital fossa or ulnar sulcus, shoulder, and Erb's point [8–10,13]. To establish neural transmission peripherally or at ventral roots in the cervical spine, M-wave- and F-wave-evoked potential recordings were recorded on the APB and ADM muscles, respectively. Parameters of amplitudes (µV) and latencies (ms) of M-wave potentials were compared between the results obtained from shooters and healthy volunteers. In the case of F-wave recordings, a frequency of no less than 14 to 20 M responses after applying electrical stimuli was analyzed and was considered proper [17]. All patients underwent sensory conduction studies (SCV) (Figure 2B) on the median and ulnar nerves after stimulation of the fingertips or the fifth finger with recordings of the prewrist areas. Pairs of surface electrodes recorded evoked potentials; the same types of surface electrodes were also used for sEMG recording. Recordings were made at an amplification of 5–5000 µV and a time base of 2–10 ms, and then the normative values recorded in healthy volunteers were compared with the values of shooters.

Motor-evoked potentials (MEPs, Figure 2C) were recorded bilaterally on the DEL, BB, APB, and ADM muscles with sEMG electrodes after stimulation of the motor neurons and levels C5–C8 of the spinal roots with a magnetic stimulus. It was delivered via a C-100 circular coil from a MagPro R30 stimulator at 70% of the maximal stimulus output (1.7 T). Recordings were made at an amplification of 100–5000 µV and a time base of 5–20 ms and were compared with normative values recorded in healthy volunteers. The output measures of the results were the amplitude (in µV) and latency (in ms) parameters of MEPs. The principles of efferent conduction of impulses from the spinal cord directly to the effector have been described elsewhere [7,10].

### 2.3. Statistical Analysis

Data were analyzed with Statistica software version 13.1 (StatSoft, Kraków, Poland). Descriptive statistics were reported as minimal and maximal values (range), mean or median values, and standard deviations (SD). The Shapiro–Wilk test was used to assess the normality of distributions in the test score. Friedman's repeated measure analysis of variance by ranks (Friedman's ANOVA) was used to determine whether there were any differences between the measurements performed on shooters and healthy subjects. *p*-values of less than 0.05 were considered statistically significant. The statistical software was used to determine the required sample size using the primary outcome variable of sEMG amplitude recorded on APB and BIC muscles during maximal contraction with a

power of 80% and a significance level of 0.05 (two-tailed). The mean and standard deviation (SD) were calculated using the data from the first five subjects in the pilot study. The sample size software estimated that at least 9 subjects were needed for this study.

## 3. Results

In a clinical study, we positively screened 13 out of 50 shooters for CTS (Figure 1). Interestingly, these people had symptoms in the shooting hand. In addition, we found two positive Froment's signs diagnosing ulnar neuropathy. The population of nine shooters did not report significant pain symptoms (VAS—scale 1), but all the participants in the vFf study showed significant hyperalgesia (Table 1). Quantitatively, the neurophysiological examination confirmed CTS in the shooting hand of four shooters (4/42, 9.5%); all the subjects had bilateral brachial plexus pathologies, and two had ulnar neuropathy of the wrist on the shooting side. In quantitative terms (Table 2), the statistical analysis showed a significant increase in the muscle tension of all the muscle groups tested, both in the proximal and distal parts of the symptomatic extremity, expressed by an increase in the amplitude parameters of more than 20–25 $\mu$V in the rEMG records (Figure 3, highest traces).

**Table 2.** Results from neurophysiological studies in healthy subjects (N = 9) and shooters (N = 9). Median or mean values with standard deviations are presented. *p*—significant differences are marked bold.

| Site of Recording | | Parameter | Control | | Shooters | | *p* | |
|---|---|---|---|---|---|---|---|---|
| | | | Right | Left | Right | Left | Right | Left |
| sEMG | DEL | rEMG ($\mu$V) | 13.3 ± 4.3 | 15.0 ± 3.5 | 18.9 ± 5.4 | 18.3 ± 4.3 | **0.05** | 0.06 |
| | | mcEMG ($\mu$V) | 1522.2 ± 198.6 | 1567.0 ± 163.6 | 1211.1 ± 132.6 | 1522.0 ± 443.8 | **0.03** | 0.13 |
| | | FE | 2.3 ± 0.5 | 2.3 ± 0.5 | 2.7 ± 0.4 | 2.3 ± 0.5 | 0.06 | 0.12 |
| | BB | rEMG ($\mu$V) | 13.8 ± 3.3 | 16.1 ± 3.3 | 22.2 ± 6.6 | 18.3 ± 6.6 | **0.02** | 0.07 |
| | | mcEMG ($\mu$V) | 1522.2 ± 454.0 | 1577.7 ± 484.1 | 1422.2 ± 840.8 | 1344.4 ± 769.9 | 0.09 | 0.1 |
| | | FE | 2.7 ± 0.4 | 2.7 ± 0.4 | 2.3 ± 0.5 | 2.2 ± 0.4 | **0.05** | **0.04** |
| | APB | rEMG ($\mu$V) | 13.9 ± 3.3 | 16.1 ± 3.3 | 18.9 ± 6.0 | 17.2 ± 7.5 | **0.02** | **0.1** |
| | | mcEMG ($\mu$V) | 1588.9 ± 459.4 | 1611.1 ± 428.4 | 1600.0 ± 107.2 | 1677.8 ± 902.4 | 0.1 | 0.12 |
| | | FE | 2.8 ± 0.3 | 2.9 ± 0.3 | 2.6 ± 0.5 | 2.5 ± 0.5 | 0.12 | 0.11 |
| | ADM | rEMG ($\mu$V) | 14.9 ± 2.5 | 15.0 ± 3.5 | 18.3 ± 5.6 | 17.7 ± 6.6 | **0.04** | 0.06 |
| | | mcEMG ($\mu$V) | 1922.2 ± 120.4 | 2355.5 ± 111.7 | 1511.1 ± 459.4 | 1688.8 ± 348.0 | **0.02** | **0.03** |
| | | FE | 2.8 ± 0.3 | 2.7 ± 0.4 | 2.7 ± 0.4 | 2.5 ± 0.5 | 0.07 | 0.07 |
| ENG—median nerve | M-wave Wrist | Amplitude ($\mu$V) | 11,111.1 ± 2204.7 | 11,111.2 ± 2227.6 | 8833.3 ± 3142.4 | 9888.9 ± 2027.6 | **0.02** | **0.03** |
| | | Latency (ms) | 3.0 ± 0.3 | 3.1 ± 0.2 | 3.6 ± 1.2 | 3.5 ± 0.9 | **0.02** | **0.05** |
| | M-wave FOC | Amplitude ($\mu$V) | 977.7 ± 1487.3 | 972.8 ± 1394.4 | 7833.3 ± 3500.0 | 8333.3 ± 2397.9 | **0.02** | **0.05** |
| | | Latency (ms) | 6.4 ± 0.6 | 6.5 ± 0.5 | 7.1 ± 0.8 | 7.2 ± 0.6 | 0.06 | 0.07 |
| | M-wave Arm | Amplitude ($\mu$V) | 11,222.0 ± 1986.1 | 11,111.0 ± 1900.3 | 9444.4 ± 4245.9 | 11,278.0 ± 2796.3 | **0.02** | 0.06 |
| | | Latency (ms) | 8.3 ± 0.4 | 8.4 ± 0.4 | 8.5 ± 1.0 | 9.0 ± 0.8 | 0.06 | 0.07 |
| | M-wave Erb | Amplitude ($\mu$V) | 11,122.0 ± 1986.1 | 11,000.0 ± 2000.0 | 10,111.1 ± 669.8 | 12,811.1 ± 3206.0 | 0.07 | 0.06 |
| | | Latency (ms) | 10.7 ± 1.7 | 10.5 ± 1.0 | 13.5 ± 0.7 | 13.4 ± 0.9 | **0.02** | **0.03** |
| | SCV—wrist | Amplitude ($\mu$V) | 16.2 ± 6.8 | 14.7 ± 7.7 | 14.0 ± 10.0 | 11.5 ± 5.3 | **0.05** | **0.03** |
| | | Latency (ms) | 2.9 ± 0.1 | 2.9 ± 0.2 | 3.2 ± 0.6 | 3.3 ± 0.4 | 0.06 | 0.07 |
| | F-wave (x/20) | | 17.8 ± 1.8 | 17.3 ± 2.0 | 15.6 ± 3.4 | 16.1 ± 3.4 | **0.05** | 0.13 |
| | M-F latency (ms) | | 22.7 ± 2.0 | 23.9 ± 2.7 | 23.8 ± 2.7 | 24.6 ± 1.9 | 0.07 | 0.06 |

**Table 2.** *Cont.*

| Site of Recording | | Parameter | Control | | Shooters | | *p* | |
|---|---|---|---|---|---|---|---|---|
| | | | Right | Left | Right | Left | Right | Left |
| ENG—ulnar nerve | M-wave Wrist | Amplitude (µV) | 10,556.0 ± 1130.4 | 10,888.9 ± 1763.8 | 9777.8 ± 322.1 | 10,111.1 ± 2204.8 | **0.05** | 0.06 |
| | | Latency (ms) | 3.1 ± 0.3 | 3.1 ± 0.1 | 2.7 ± 0.5 | 2.7 ± 0.2 | 0.06 | 0.07 |
| | M-wave Sulcus uln. | Amplitude (µV) | 9000 ± 866.3 | 9666.7 ± 866.0 | 7477.8 ± 3362.2 | 8666.7 ± 1145.6 | **0.05** | 0.06 |
| | | Latency (ms) | 6.4 ± 0.4 | 6.5 ± 0.4 | 6.7 ± 0.5 | 6.5 ± 0.6 | 0.1 | 0.1 |
| | M-wave Arm | Amplitude (µV) | 9888.9 ± 21453.0 | 9777.8 ± 1201.9 | 9666.7 ± 3741.7 | 9000.0 ± 1500.0 | 0.1 | **0.05** |
| | | Latency (ms) | 8.2 ± 0.5 | 8.3 ± 0.3 | 8.8 ± 0.9 | 8.9 ± 0.9 | 0.1 | 0.1 |
| | M-wave Erb | Amplitude (µV) | 10,556.0 ± 1130.4 | 10,889.0 ± 1763.8 | 4844.4 ± 3644.4 | 3644.4 ± 3178.1 | **0.03** | **0.02** |
| | | Latency (ms) | 10.5 ± 1.6 | 10.4 ± 1.2 | 13.1 ± 1.2 | 12.9 ± 0.7 | **0.03** | **0.02** |
| | SCV—wrist | Amplitude (µV) | 16.7 ± 4.3 | 16.1 ± 6.4 | 10.9 ± 4.6 | 10.8 ± 4.3 | **0.02** | **0.02** |
| | | Latency (ms) | 2.9 ± 0.2 | 2.9 ± 0.1 | 2.7 ± 0.3 | 3.0 ± 0.5 | 0.07 | 0.06 |
| | F-wave (x/20) | | 18.1 ± 1.5 | 18.1 ± 1.5 | 17.3 ± 2.6 | 16.7 ± 2.4 | 0.06 | 0.07 |
| | M-F latency (ms) | | 25.3 ± 0.7 | 25.5 ± 0.8 | 23.9 ± 2.2 | 25.0 ± 2.1 | 0.06 | 0.07 |
| MEP | DEL | Amplitude (µV) | 1188.9 ± 169.1 | 1177.8 ± 120.1 | 477.8 ± 192.2 | 700.0 ± 531.1 | **0.02** | **0.03** |
| | | Latency (ms) | 4.9 ± 0.3 | 5.1 ± 0.3 | 4.9 ± 0.3 | 5.1 ± 0.3 | 0.1 | 0.1 |
| | BB | Amplitude (µV) | 1166.7 ± 122.5 | 1188.9 ± 161.6 | 1066.7 ± 878.9 | 1311.1 ± 873.9 | 0.1 | 0.07 |
| | | Latency (ms) | 5.5 ± 0.4 | 5.6 ± 0.4 | 5.9 ± 0.8 | 6.1 ± 0.6 | **0.02** | **0.03** |
| | APB | Amplitude (µV) | 1200.0 ± 165.8 | 1200.0 ± 200.0 | 866.6 ± 878.9 | 511.1 ± 275.8 | **0.03** | **0.02** |
| | | Latency (ms) | 13.6 ± 0.3 | 13.5 ± 0.4 | 14.1 ± 1.1 | 14.0 ± 0.9 | 0.1 | 0.1 |
| | ADM | Amplitude (µV) | 1211.1 ± 161.5 | 1155.6 ± 113.4 | 677.8 ± 108.3 | 644.4 ± 339.5 | **0.03** | **0.02** |
| | | Latency (ms) | 13.2 ± 0.5 | 13.4 ± 0.5 | 14.2 ± 1.6 | 14.5 ± 0.9 | 0.09 | 0.1 |

Abbreviations: rEMG—amplitude at rest, mcEMG—amplitude during maximal contraction, FE-modified frequency index (3–0)—frequency of motor unit action potential recruitment during maximal contraction sEMG recording (3—95–70 Hz, 2—65–40 Hz, 1—35–10 Hz, 0—no contraction), DEL—deltoid muscle, BB—biceps brachii muscle, APB—abductor pollicis brevis, ADM—abductor digiti minimi muscle, sEMG—surface electromyography, ENG—electroneuronography, SCV—sensory conduction studies, and MEP—motor-evoked potential induced transvertebrally at cervical level.

A simultaneous decrease in the activity of the motor units tested during maximal contraction was manifested in the shooters as a significant decrease in the mean amplitude parameter in the mcEMG recordings (compared to the normative results of the healthy volunteers). The electroneurographic recordings of nerve impulse conduction in the sensory and motor nerve fibers showed a significant decrease in the amplitude parameter in the M- and SCV-wave recordings of the median nerve potentials. This indicated the consequences of carpal tunnel syndrome and generalized neuropathy of the motor fibers of the ulnar nerve along its entire length with a predilection to the level of the wrist (Figure 3, ENG results). The results of the F-wave examination showed slight abnormalities in the conduction of nerve impulses in the motor fibers of the ventral roots at the C6–C7 level on the shooting side (a decrease in the F-wave frequency potential following median nerve stimulation). The highest sensitivity in detecting a sequence of abnormalities caused by shooting was shown in the studies of MEP efferent transmission from the level of cervical motor neuromeres C4–C8 to the effectors on the shooting side; it was expressed by a decrease in amplitude (in the DEL, APB, and ADM recordings; Figure 3, the lowermost recordings) and a latency increase (in the BB recording; Table 2).

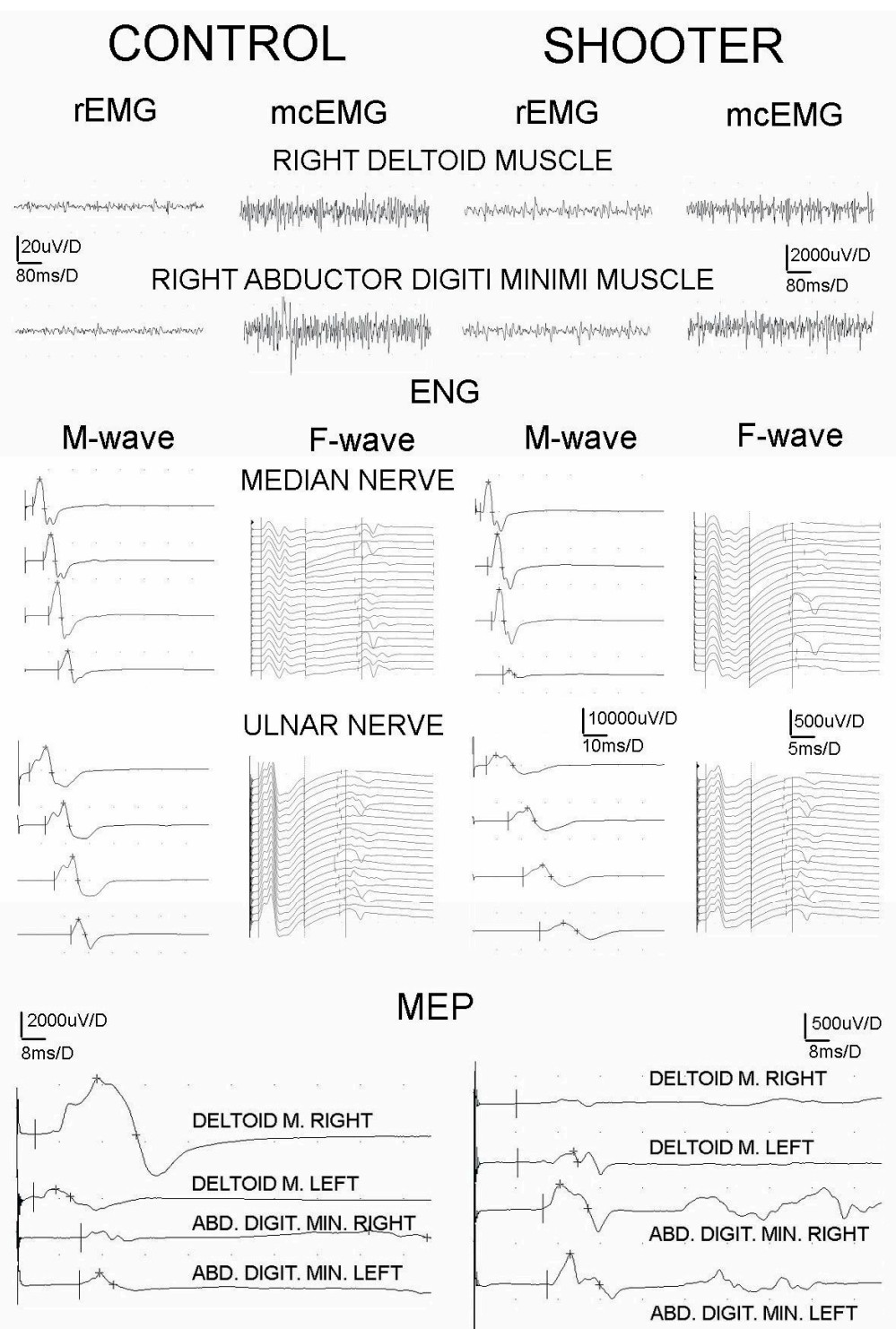

**Figure 3.** Examples of sEMG, ENG, MEP recordings of healthy volunteer and shooter for comparison. Note different amplification of MEP recordings; they were four times more amplified in the case of the shooter as indicated by the numbers close to the vertical bars. Abbreviations: rEMG—amplitude at rest, mcEMG—amplitude during maximal contraction, ENG—electroneuronography, and MEP—motor-evoked potential induced transvertebrally at cervical level.

## 4. Discussion

This study evidences the development of functional changes in the peripheral neuronal conduction of the motor and sensory fibers in the nerves of the upper extremities in shooters following six months of regular training. This conclusion is drawn from the results of the

M-wave ENG exams. In addition, the studies of antidromic stimulation in the ENG tests also revealed subclinical signs of the consequences of C6–C7 disc–root conflicts. These pathologies were accompanied by increased muscle tension and decreased activity in the motor units in both the proximal and distal muscles in the upper extremities of the shooters, which were found on the basis of sEMG recordings. The above-mentioned abnormalities resulted in a decrease in the amplitude parameters of the MEP recordings in relation to the reference values determined with the healthy controls, which indicated a change in the total efferent transmission from the cervical motor centers to the effectors. The graph in Figure 4 summarizes the main results of this study. The question arises whether the above-mentioned pathologies could have been present in the subjects before the required recruitment for this study, the 6-month shooting training period. However, it should be remembered that, during the screening clinical trials, the subjects could not report any subjective symptoms (even subclinical ones), confirming the development of upper limb pathologies or cervical pain.

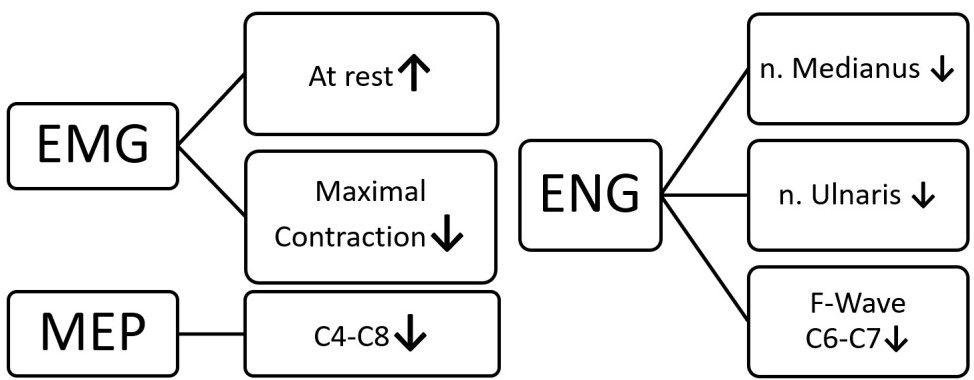

**Figure 4.** Summary of the main results of the presented study.

Interestingly, two shooters developed CTS in both hands, and all suffered from brachial plexus neuropathies on both sides, suggesting that the recoil hit at the wrist is not the only cause. Holding a shooting position for a long time probably leads to such results. This claim can be reinforced by the results of studies in which shooters suffer from low back pain as often as weightlifters, proving that this discipline significantly affects the tissues at the level of the spine [18]. Moreover, Ojanen et al. [19] found a positive correlation between shooting scores in the standing position and abnormalities in the muscular strength of the lower limbs and upper body. These results were displayed in military field training; however, it can also be assumed that sport shooters, in order to achieve high accuracy, must increase the strength of their postural muscles and overload their tissues at the level of the spine. However, we cannot rule out that the extended arm can transmit sufficient recoil force to the proximal levels of the upper limb and hit the nerves on the opposite side as well.

There is also the possibility of the long-term impact of the inorganic lead fumes produced by shooting, due to their neurotoxic properties, affecting the development of neuropathy [20]. The most plausible explanation for our results involves the combined effect of all these factors. However, the latter is least likely, as the shooting range of the clubbers seemed to be properly ventilated.

The knowledge provided in this study can be used in clinical practice by sports and military physicians as well as general practitioners to detect neuropathy before it leads to serious complications and thus a reduction in the quality of life. Understanding that shooters are at a higher risk of this type of injury, it would be prudent to find ways to mitigate the nerve effects associated with this sport with innovative technologies; the creation of ergonomic workplaces and the strengthening of healthy postures in everyday life should be promoted. In addition, it is possible that different types of handles on handguns differ in their impact on the user; therefore, the creation of the most ergonomic one may

be beneficial in the context of the prevention of upper limb neuropathy. Interestingly, during military field training, the score for shooting a rifle in the standing position was reduced in the midtraining measurements, while the score obtained in the prone position did not change [19]. Shooters should be recommended to maintain a rigorous time interval between successive training instead of intensive training without breaks, which affects the development of functional changes in the brachial plexus fibers and muscles. The use of soft shooting gloves and orthoses strengthening the wrist may be of a protective nature to prevent the effects of recoil. The standing position may exert significant stress on the body, and thus the alleviation of this effect should be achieved by training in shooting sports in the prone position. The abovementioned study did not assess the neurophysiological properties of the nerves of the upper extremities, and, therefore, its application is limited in comparison to the results obtained in our research.

The majority of papers on the reasons for injuries of the brachial plexus and the consequences of trauma in sports are mainly related to accidents in many disciplines and are not a description of the long consequences of shooting [21], which is a novelty in our report. The precision of the concentric needle EMG recordings in comparison to the surface EMG recordings during the detection of axonal neuropathies coexisting with brachial plexus injuries is indisputable [22]. However, the noninvasive approach, especially in correlation with the results of the MEP recordings also recorded with the surface electrodes, appears to be more and more appreciated [23]. Our study is an example, especially of the subjects constituting the healthy control group and the select number of shooters. The precision of the proposed semiautomatic analysis of the sEMG recording parameters becomes important in screening tests where discrete neurogenic effects are expected [10].

We are aware of some limitations of our study: (1) The study group was relatively small and heterogeneous; therefore, our results may, to some extent, be due to differences in anthropological characteristics or lifestyle. (2) Shooters mainly used handguns; our results do not fully apply to users of longer firearms. (3) Shooters used handguns of various weights, handles, calibers, and types of ammunition (side and central), which may affect their impact on the body function, especially on the muscle and nervous systems. Our preliminary study focused on confirming or refuting the diagnosis of CTS, as our screening process targeted this pathology. However, due to the phenomenon of "double crush syndrome" and the biomechanics of shooting, which cause the propagation of the recoil force after the shot to the entire upper extremity, we concluded that investigating whether this sport activity may contribute to non-CTS neuropathies was an innovative concept.

Considering the observation period in our study, in fact, it was longer in most cases than six months; however, we considered the regularity and continuity of shooting training in the subjects we studied, and it was the main condition that had to be fulfilled for the persons screened preliminarily. For this reason, we acclaimed the six-month observation period in order for it to be the shortest to be considered as the regular performance of the sport.

We admit that the real percentage of CTS in bullseye shooters might be higher than we established. However, we used the screening design comprising three sensitive clinical tests from which one positive result to qualify the subjects for neurophysiological evaluation was sufficient enough to reach a reliable detection rate. The highly sensitive EMG and ENG neurophysiological tests did not display the incidence of CTS in the control group with demographic and anthropometric properties similar to those of the population of shooters in this study. The 9.5% CTS incidence had to be caused by an important harmful factor in bullseye shooters, and the recoil can be mentioned as the first suspected factor. The possible effect of the rapid and repetitive recoil could have evoked nerve fiber traction in both the proximal and distal parts of the brachial plexus on the shooting side, causing the axonal changes detected in the ENG and MEP studies.

### 5. Conclusions

The increased incidence of CTS and brachial plexus pathologies, accompanied by statistically significant abnormalities in the presented results of the neurophysiological examinations of the shooters, indicates that this activity is a risk factor for their development. The most likely explanation is that nerve fiber traction is often associated with sudden brachial plexus injuries, leading to axonal-type (axon loss) pathologies. A better understanding of the various risk factors associated with shooting sports will be achieved by conducting large multicenter studies in the future on homogeneous populations in terms of equipment, training, BMI, age, and gender.

**Author Contributions:** Conceptualization, A.R. and J.H.; methodology, A.R. and J.H.; validation, A.R. and J.H.; formal analysis, J.H.; investigation, A.R., P.D., A.F., K.B., M.K. and J.H.; resources, P.D. and J.H.; data curation, A.R. and J.H.; writing—original draft preparation, A.R., A.F., K.B., M.K. and J.H.; writing—review and editing, A.R., P.D., A.F., K.B., M.K. and J.H.; visualization, A.R. and J.H.; supervision, J.H.; project administration, A.R. and J.H.; funding acquisition, P.D. and J.H. All authors have read and agreed to the published version of the manuscript.

**Funding:** This research received no external funding.

**Institutional Review Board Statement:** This study was conducted according to the guidelines of the Declaration of Helsinki and was approved by the Bioethics Committee of the University of Medical Sciences (decision No 554/2017) for studies involving human.

**Informed Consent Statement:** Written informed consent was obtained from all the subjects involved in the study.

**Data Availability Statement:** All the data generated or analyzed during this study are included in the published article.

**Conflicts of Interest:** The authors declare that there is no conflict of interest.

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
