# Peer review of "Incidence of Carpal Tunnel Syndrome and Other Coexisting Brachial Plexus Neuropathies in Bullseye Shooters—A Pilot Retrospective Clinical and Neurophysiological Assessment"

_applsci, doi:10.3390/app13148020_

Round 1
Reviewer 1 Report
Line 17 (abstract): surface should be added to the electromyography
Line 29 (keywords): please add surface EMG and MEP recordings
Line 223 (Figure 3): please add the remark that the scales for controls and shooters are different
Line 239: Latencies for biceps brachii muscle are not presented in Figure 3 but in Table 2, please modify appropriately
Discussion: please compare and discuss the potential of surface EMG and concentric EMG in detecting axonal neuropathies of (brachial) plexus
Additional comments:
The main question addressed by the research was if the shooting sport with the handgun by civils is the risk factor for carpal tunnel syndrome and other neuropathies in upper limb.
The study addresses a specific gap in the field regarding the methodology used, since other studies did not use neurophysiological assessments to investigate nerves of upper extremity in shooters.
The methodology used and the controls used are appropriate.
The conclusions are consistent with the evidence and arguments presented.
The references are appropriate.
Tables and figures are adequate.
Author Response
Please see the attachement

Reviewer 2 Report
#1
This study was not a prevalence study of bullseye shooters. This is a pilot study or a preliminary study. And the authors discussed carpal tunnel syndrome as well as other upper arm neuropathy. The prevalence of CTS in shooters was not clearly discussed in the manuscript.
#2
The authors used ‘qualitative’ for EMG analysis, but I think the analysis should be regarded as a quantitative analysis as it used a consecutive number for analysis)
#3
The authors did not confirm EMG for the whole shooter group. I think someone in the excluded people and symptom-free people might have positive EMG results. The prevalence of CTS or other upper arm neuropathy may be higher if the authors included all members of the shooter group.
#4
In the general population, the prevalence of carpal tunnel syndrome is known to be around 5%. Although this shooter group showed 9% of incidence of carpal tunnel syndrome, the higher incidence might be affected by several factors such as highly sensitive EMG, selection bias (healthy people tend to play more active sports). Thus this result cannot reveal a higher incidence of CTS in shooter group and the negative effect of the shooting on the nerves.
Thus, I think this manuscript cannot suffice the requirement of publication.
Reviewer 3 Report
The importance of the manuscript should be emphasized. Line 64-67.
The null hypothesis should be used instead of the zero hypothesis. Line 66.
Is a minimum of 6 months enough? What is the answer to this question? Line 79
NS is written in Table 1. P value should be given. p=0.50 and p=0.90 are not the same expression. Therefore, the value should be written in numbers. Line 96
Since there are two groups and the groups are dependent, a repeated measures ANOVA test should also be performed. Line 181.
What do the authors recommend to shooters to avoid injuries? Line 304.
Four references were cited in the discussion. Discussion should be improved.
Moderate editing of English language.
Author Response
Please see the attachement

Round 2
Reviewer 2 Report
Still, the inclusion criteria was "no subjective symptoms" and some of the symptom free people were screened positive for CTS in the interview. I think that was also a paradox.
There was no neurologic examination for brachial plexus or thoracic outlet syndrome. Some patients might not be included.
In conclusion, nerve fiber traction was not discussed in this manuscript at all. I think this could be moved into the discussion.
Reviewer 3 Report
Corrections have been made. I think your manuscript will be read more with these fixes.
Minor editing of English language required
